# Biodegradable Food Packaging of Wild Rocket (*Diplotaxis tenuifolia* L. [DC.]) and Sea Fennel (*Crithmum maritimum* L.) Grown in a Cascade Cropping System for Short Food Supply Chain

Perla A. Gómez [1,*], Catalina Egea-Gilabert [1,2], Almudena Giménez [2], Rachida Rania Benaissa [2], Fabio Amoruso [2], Angelo Signore [2,3], Victor M. Gallegos-Cedillo [2,4], Jesús Ochoa [1,2] and Juan A. Fernández [1,2,*]

1   Institute of Plant Biotechnology, Universidad Politécnica de Cartagena, 30202 Cartagena, Spain; catalina.egea@upct.es (C.E.-G.); jesus.ochoa@upct.es (J.O.)
2   Department of Agronomical Engineering, Universidad Politécnica de Cartagena, 30203 Cartagena, Spain; almudena.gimenez@upct.es (A.G.); rachidarania.benaissa@edu.upct.es (R.R.B.); fabio.amoruso@upct.es (F.A.); angelo.signore@upct.es (A.S.); victor.gallegos@upct.es (V.M.G.-C.)
3   Department of Soil, Plants and Food Sciences, University of Bari Aldo Moro, 70126 Bari, Italy
4   Department of Engineering, CIAIMBITAL Research Centre, University of Almería, 04120 La Cañada, Spain
*   Correspondence: perla.gomez@upct.es (P.A.G.); juan.fernandez@upct.es (J.A.F.)

**Abstract:** The environmental impact of food products is significantly affected by their packaging. Therefore, this study aimed to assess the effect of PLA (polylactic acid) film, as an alternative to petroleum-based bags, on the shelf-life of fresh-cut wild rocket and sea fennel grown in a cascade cropping system (CCS). To this end, wild rocket (main crop) was cultivated using either peat or compost as a growing medium. Sea fennel (secondary crop) was subsequently grown in a floating system with leachates from the primary crop as a nutrient solution. The leaves of both crops were harvested and packaged in OPP- (oriented polypropylene) or PLA-based bags and stored for 7 days at 4 °C. The leaves of wild rocket and sea fennel showed lower dehydration and lower respiration when compost was used as a growing medium or leachate. Wild rocket in compost increased in nitrate and vitamin C contents at harvest while leachates had scarce influence on their contents in sea fennel. After storage, regardless of the crop, no relevant detrimental changes were observed on leaves packaged with PLA, being a product microbiologically safer when compared to OPP. The bag type had almost no influence on most relevant phytochemical compounds. In conclusion, the use of a PLA-based film on minimally processed wild rocket and sea fennel leaves is a sustainable alternative to petroleum-based plastic for a short food supply chain.

**Keywords:** sustainable agriculture; growing media; fresh-cut; postharvest; compostable package; food distribution; phytochemical compound

## 1. Introduction

Nowadays, innovative urban and peri-urban plant production systems, especially in developed economies, are gaining popularity [1], particularly those that can increase resource efficiency [2], which achieves ecosystem benefits and mitigates climate change [3]. The enlarging of the supply chain that has developed over the years has produced effects from ecological, economic, and social points of view. Thus, some years ago, the short food supply chain appeared as a new concept and has expanded significantly, particularly in European countries, as it is supported by EU policies [4]. These production systems with increased circularity can both enhance food security in cities and reduce the environmental impact that results from long transportation distances, which involve elevated energy consumption for refrigeration and the use of chemical refrigerants [5].

A cascade cropping system (CCS) that collect the leachate of a substrate from a main crop to cultivate a secondary crop is of particular interest in naturally dry and nutrient-poor environments [6]. Closing the nutrient cycle in soilless systems increases resource efficiency by reducing fertiliser and water consumption [7]. In addition, any reduction in the environmental impact of agriculture also translates into a clear interest from consumers [8].

The search for organic materials that can be used as peat alternatives has become increasingly important, with the compost from agricultural and agri-food industry waste being an option as far as it is considered environmentally friendly and in agreement with the circular economy. Furthermore, compost can be an important source of biofertilisation and bio-stimulation of crops because of its nutritional elements, humic substances, and hormone-like molecules released by microorganisms [9]. Additionally, composts made from specific agro-industrial wastes have been proven to suppress a wide variety of soil-borne plant pathogens, including some oomycetes in baby leaf vegetables [10].

The commercialisation of leafy vegetables, including baby leaf vegetables and micro-greens, is of increasing interest, offering customers convenient products rich in healthy bioactive compounds; one example is wild rocket (*Diplotaxis tenuifolia* (L.) DC.), an important component of ready-to-eat salads [11]. In addition to its distinct taste and peppery flavour, the potential health benefits associated with leafy phytochemicals, such as flavonoids and glucosinolates, have been recently addressed [12]. Likewise, but to a lesser extent, sea fennel (*Crithmum maritimum* L.), widely distributed throughout the Mediterranean and Atlantic seacoasts [13], has generated interest due to its desirable organoleptic attributes and health-promoting properties, including a relatively high content of fatty acids, predominantly linolenic and linoleic acids, vitamin C, and carotenoids [14,15]. Fresh leaves of this plant are versatile and can be consumed in various ways, including in salads, soups, sauces, and as a spice, particularly in fish-based dishes [16].

One of the most interesting issues concerning fresh food products is packaging, as this is one of the major factors that contribute to the overall environmental impact of products and represents one of the greatest variables affecting the sustainability of the supply chain [17]. Leafy vegetables are typical of the fresh-cut industry with petroleum-based films (e.g., polypropylene) commonly used for their packaging. However, these materials are related to global warming, ozone depletion, and non-renewable energy [18]. Millions of tons of nondegradable packages ultimately end up deposited in landfills, even when the worldwide recycling rates are increasing.

To reduce the environmental impact of plastic bags, polylactic acid (PLA) bags are being used as substitutes for packaging fresh produce due to their properties, even when they present some limitations that might prevent them from being competitive with conventional plastics [19]. PLA is a kind of biodegradable thermoplastic polymer, recyclable, compostable, produced from renewable sources (mainly from starch feedstocks), and approved for contact with food [18]. PLA is a versatile material, being thermo-sealed, a gas barrier, UV resistant, biocompatible, elastic, rigid, and hydrophobic [20,21]. PLA can retain the beneficial material characteristics of petrochemical-based packages while allowing for a transition towards a circular economy by reducing fossil resource usage. The current bioplastic market accounts for less than 1% of the entire plastic packaging market [22]. However, before large-scale systemic changes are adopted, full environmental evaluations should be considered. The physical and mechanical properties of PLA-based packages are continuously improved to meet the requirements of different commodities. Composites based on biodegradable compounds can be helpful for preserving freshness and retarding microbial spoilage. Recent research has demonstrated that porous biodegradable sodium alginate composite fortified with *Hibiscus sabdariffa* L. was appropriate for extending the shelf-life of highly perishable climacteric fruits [23]. It can be seen as a part of hurdle technology, which combines preservative techniques with synergistic effects to reduce losses and maintain nutritional value. Among the tailored PLA-based compositions, PLA coated with Kraft paper can enhance the barrier to water and air to replace non-biodegradable polymers. Nevertheless, the effectiveness of PLA in prolonging the shelf-life of fresh

salads has been scarcely analysed [24], as the possibility of storage for long periods is still unknown. Short food supply chains for vegetables open the possibility of selecting compostable and biodegradable films for modified atmosphere packaging (MAP), with foreseeable good results.

The purpose of this study was to evaluate the postharvest life of wild rocket (main crop) cultivated in peat or compost, and sea fennel cultivated in a floating system (secondary crop) using a CCS. Each vegetable was separately packaged in a PLA-based film as an alternative to petroleum-based bags for a short food supply chain. To our knowledge, there are not any previous reports evaluating a biodegradable package for the short food supply chain of leafy vegetables grown in a CCS.

## 2. Materials and Methods

### 2.1. Plant Material and Growing Conditions for Main Crop (Wild Rocket)

The characteristics of the main crop (wild rocket [*Diplotaxis tenuifolia* (L.) DC. cv. Apollo]) are described by Signore et al. [25]. Briefly, plants were cultivated in metal gutters filled with peat or an agro-industrial compost as growing media. A mix of white/black (60/40—in volume) peat was used, while the compost was composed of tomato and pepper juice waste, leek waste, and vineyard residues in dry weight percentages of 41, 43, and 16%, respectively. Fertigation was done daily with an automated system, using a nutrient solution with the following (in mM): 7.2 $NO_3^-$, 4.8 $NH_4^+$, 2 $H_2PO_4^-$, 2.5 $SO_4^{2-}$, 6 $K^+$, 1.9 $Ca_2^+$, and 1.5 $Mg^{2+}$, while micronutrients and iron were provided as a commercial solution: Nutromix® (2 mg $L^{-1}$ for microelements—Biagro, Massalfassar, Valencia, Spain) and Sequestrene® G100 Syngenta (7% soluble Fe, 6% chelated Fe, 1.5 mg $L^{-1}$—Basel, Switzerland).

Harvest was performed when rocket plants had seven to eight leaves, corresponding to their appropriate commercial stage. Two harvests were carried out, 27 and 55 days after transplanting, respectively. The data presented in this work belong to the second harvest, as those from the first harvest were not significantly different from and were consistent with those from the second.

### 2.2. Plant Material and Growing Conditions for Secondary Crop (Sea Fennel)

Sea fennel (*Crithmum maritimum* L., Semillas Cantueso, Dunas de Artola, Málaga) seeds were sown on 4 May 2022, in styrofloat trays (0.6 m × 0.41 m) filled with peat, in a growth chamber at 20 °C for 5 days and then transferred to floating beds (1.35 × 1.25 × 0.2—L × W × H, respectively), floating on fresh tap water with an electrical conductivity (EC) of 1.1 dS $m^{-1}$ and a pH of 7.8. Aeration was provided using a blow pump connected to a pipe trellis positioned at the bottom of each flotation bed. On 25 May 2022, thinning was done to reach a final plant density of 256 plants $m^{-2}$. The tap water of each bed was replaced, one month after sowing, with 200 L of the following treatments: 'Peat leach' = 100% peat leachate (pH 7.9, EC 3.6 dS $m^{-1}$), 'Compost leach' (pH 7.7, EC 3.1 dS $m^{-1}$) = 100% compost leachate (both collected from wild rocket crop), and 'Control' = 100% NS on which the concentration of every nutrient was 0.6 strength of that used for rocket crop cultivation (pH 7, EC 2.3 dS $m^{-1}$), in order to ensure comparability with previous results [26]. The average DLI during sea fennel cultivation was 14.9 mol $m^{-2}$ $d^{-1}$, while the lowest, highest, and average air temperatures inside the greenhouse were 14.0, 43.2, and 27.1 °C, respectively. On 18 July when the plants had 3–4 true leaves, the harvest was carried out.

### 2.3. Processing, Packaging and Storage

After harvest, the plant material (wild rocket and sea fennel leaves) was minimally processed in a disinfected cold room at 8 °C. Leaves free of defects were sanitised by immersion for 1 min in a solution of chlorinated water (150 ppm NaOCl, pH 6.5, 4 °C) and then rinsed for 1 min in tap water (4 °C) to ensure that the final chlorine residue was below 5 ppm. After being drained for 10 min in a perforated basket, samples of about 100 g were arranged for passive MAP in 30 cm × 20 cm × 35 μm thick OPP (oriented polypropylene) plastic bags (Plásticos del Segura, Murcia, Spain) or in 30 cm × 20 cm, 20 μm thick PLA

bags. The PLA bags had one side made of transparent PLA (42.5%) and on the other side, the PLA was coated with an eco-layer of Kraft paper (57.5%) (Classpack-Nativia® NTSS, Barcelona, Spain). The transmission rates of $O_2$ and $CO_2$ at 23 °C and 0% relative humidity were similar for OPP bags, with a value of 11,000 $cm^3$ $m^{-2}$ $d^{-1}$ $atm^{-1}$ for both gases. In contrast, for PLA bags, the transmission rates were ten times less, with a value of 1100 $cm^3$ $m^{-2}$ $d^{-1}$ $atm^{-1}$ for both gases (data provided by the suppliers). For OPP and PLA, water vapour transmission rates (WVTR) were 18 g $m^{-2}$ $d^{-1}$ and 330 g $m^{-2}$ $d^{-1}$, respectively. Both types of bags were then thermally sealed on the top using a thermo-sealer (Lovero Bag Sealer-SK 410, Korea).

The acid of the PLA film was obtained from corn and sugar cane and was produced according to the standards of good industrial techniques in compliance with Regulations (EU) 1935/2004 and (EU) 10/2011. Kraft paper was obtained from 100% virgin long fibre which gives cleanliness and consistency (Smurfit Kappa, Ireland), according to Forest Stewardship Council. Global migration analysed by using modified polyphenylene oxide (MPPO) as a solid food simulant for the food contact side (testing conditions: 10 d/20 °C–22 °C) was below the limit of quantification (DIN EN 1186:2002-07/2002-12). Five replicates from each treatment (peat and compost for wild rocket, and 'Peat leach', 'Compost leach', and 'Control' for sea fennel), package (OPP and PLA), and MAP storage duration (processing day = day 0, and after 7 days = day 7) were prepared and stored in darkness at 4 °C, 90% relative humidity (RH). Storage period was set at just 7 days because the packaged produce was intended only for a short food supply chain. Three replicates were randomly selected for physicochemical analysis.

### 2.4. Physicochemical Analyses

#### 2.4.1. Head-Space Composition

On sampling day, before opening the bags, the atmosphere composition within the packages was assessed by an $O_2/CO_2$ head-space analyser (PBI-Dansensor CheckPoint, Ringsted, Denmark). The test needle of the gas analyser was inserted into each package through an adhesive silicon septum to avoid air leaking from the head-space. Data were expressed as kPa.

#### 2.4.2. Weight Loss

Weight loss was calculated as the difference between the weight of the samples at the beginning of storage and their weight after 7 days. For data standardisation, weight loss was expressed as a percentage (%) of the initial value.

#### 2.4.3. Microbial Quality

Microbial growth (mesophilic and psychrophilic aerobic bacteria, enterobacteria, and yeast and mould growth) was determined using standard enumeration methods. Samples of 1 g poured into a sterile stomacher bag (model 400 Bags 6141, London, UK) were homogenised with a 10 mL sterile peptone saline solution (pH 7; Scharlau Chemie SA, Barcelona, Spain) for 10 s in a masticator homogeniser (Colwort Stomacher 400 Lab, Seward Medical, London, UK). For the enumeration of each microbial group, 10-fold dilution series were prepared in 9 mL of sterile peptone saline solution. Mesophilic, enterobacteria, and psychrotrophic were pour-plated, and yeast and mould were spread-plated. Media (Scharlau Chemie, Barcelona, Spain) and incubation conditions were plate count modified agar (PCA) for mesophilic and psychrophilic aerobic bacteria (30 °C for 48 h and 5 °C for 7 days, respectively); violet-red bile dextrose agar for enterobacteria (37 °C, 48 h); and rose Bengal agar for yeasts and moulds (22 °C, 3–5 days). All microbial counts were reported as log colony forming units per gram of product (log CFU $g^{-1}$). Each of the three replicates was analysed by duplicate.

### 2.4.4. Colour

The colour of leaves was determined on three points at the upper side of 10 leaves from each replicate using a colourimeter (Minolta CR-400 Series, Ramsey, NJ, USA). Tristimulus parameters (L*, a*, b*) of the CIE Lab system were used to calculate the hue angle = arctan (b*/a*).

### 2.4.5. Sensory Evaluation

The sensory analysis was performed by a trained panel according to international specifications (ASTM STP 913, 1986) in a standardised room (UNE-EN ISO 8589 2007) equipped with ten testing boxes as described by Amoruso et al. [26]. The panel developed a vocabulary of sensory attributes including visual appearance, colour, dehydration, aroma, flavour, and texture. The overall quality was described as the global acceptance of the product that included visual, textural, and taste attributes. The samples were scored on a 9-point scale for colour, aroma, flavour, texture, visual appearance, and overall quality (9: excellent, 5: limit of marketability, 1: extremely bad) as well as for dehydration (9: without dehydration, 5: limit of marketability, 1: extremely dehydrated:). Sparkling mineral water was used as palate cleanser. The evaluation was done by 5 trained judges at day 0 and after 7 days of storage at 4 °C.

### 2.4.6. Nitrate Content

Nitrates were extracted in triplicate per treatment. The extraction from 0.2 g of dry leaf samples was carried out with 50 mL of distilled water by continuous agitation in an orbital shaker (Stuart SSL1, Stone, UK) for 45 min at 110 rpm at 50 °C. Nitrate concentration was determined by ion chromatography using a Metrosep A SUPP 5 column (Metrohm AG, Zofingen, Switzerland) at a flow rate of 0.7 mL min$^{-1}$, following the manufacturer's instructions.

### 2.4.7. Vitamin C

The vitamin C content was determined as the combined amount of ascorbic acid (AA) and dehydroascorbic acid (DHA), using high-pressure liquid chromatography as described by Zapata and Dufour [27] with slight modifications. Briefly, three grams of frozen samples were crushed, and 6 mL of an extraction solution of citric acid 0.1 M, 0.05% EDTA, and 4 nM NaF in 5% methanolic water were added. The mixture was homogenised for 30 s at high speed (Ultraturrax T25 basic, IKA, Germany). The homogenate was filtered through a sterile gauze with the pH being adjusted to 2.3–2.4 with 6 N HCl. The filtrate was transferred into tubes and centrifuged for 5 min at 13,500 rpm and 4 °C (Sorvall RC-SB series centrifuge). Subsequently, the samples were passed through a SepPak C18 cartridge (Waters Assoc.) and filtered again with a spin filter (0.45 µm). Then, 750 µL of the filtrate were placed in an HPLC vial and 250 µL of 1,2-o-phenylenediamine dihydrochloride (OPDA) (0.83 mg mL$^{-1}$) in methanol/water (5:95, *v/v*) were added. The mixture was allowed to react for 37 min at room temperature for derivatisation to form a fluorescent condensation product for detecting DHA. Then, 20 µL were injected on a Gemini NX C18-110 column (250 mm × 4.6 mm, 5 µm particle size; Phenomenex, Torrance CA, USA), using an HPLC system (Shimadzu, Kyoto, Japan) equipped with an SPDM-20A photodiode array detector. The mobile phase was 5 mM hexadecyl trimethyl ammonium bromide, 50 mM $KH_2PO_4$, and 5% methanol (pH 4.59) with an isocratic flow of 1.8 mL min$^{-1}$. Chromatograms were recorded for 14 min at 261 nm (AA, Rt = 6.4 min) and 348 nm (DHA, Rt = 3.1 min). AA and DHA were quantified using commercial standards (Sigma, St. Louis, MO, USA). Calibration curves for quantification were made with at least eight data points from 1.25 to 0.1 mM and from 1.25 to 0.01 mM for AA and DHA, respectively. Results were expressed as the sum of AA + DHA, as g kg$^{-1}$ fresh weight (FW).

#### 2.4.8. Total Phenolics and Total Flavonoids Content

The total phenolic content (TPC) and the total flavonoid content (TFC) were determined as described by Martínez-Zamora et al. [28]. Briefly, 19 μL of the sample extract were mixed with 29 μL of 1 N Folin–Ciocalteu reagent and 192 μL of 0.4% $Na_2CO_3$ 2% NaOH. After 1 h incubation in darkness, the absorbance was measured at 750 nm using a microplate reader (Tecan Infinite M200, Mannedorf, Switzerland). The TPC was expressed as mg of chlorogenic acid equivalents (ChAE) $kg^{-1}$ FW. Each extract was analysed in triplicate. For TFC, 30 μL of extract were mixed with 80 μL of 20 g $L^{-1}$ $AlCl_3$. After shaking and 1 h incubation in darkness, absorbance was measured at 415 nm. The TFC was expressed as mg of rutin equivalents (RE) $kg^{-1}$ FW. Each sample extract was analysed in triplicate.

#### 2.4.9. Total Antioxidant Capacity

Total antioxidant capacity (TAC) was analysed by the ferric reducing antioxidant power (FRAP) method according to Castillejo et al. [29] from the same extract prepared for TPC. A daily reaction solution containing sodium acetate buffer (pH 3.6), 10 mM (2,4,6-tripyridyl-S-triazine (TPTZ) solution (in 40 mM HCl), and 20 mM $FeCl_3$ was prepared in a proportion of 10:1:1 (volume) and incubated at 37 °C for 2 h in darkness. Then, 198 μL of FRAP solution were added to 6 μL of rocket extract and incubated for 30 min at room temperature in darkness. The TAC was measured by changes in absorbance at 593 nm (Tecan Infinite M200, Mannedorf, Switzerland). Obtained data were expressed as mg of Trolox equivalents (TE) $kg^{-1}$ FW. Each sample extract was analysed in triplicate.

#### 2.5. Experimental Design and Statistical Analyses

Experiments for the main and the secondary crop were a randomised complete block design, respectively, with three replicates ('beds'). Every bed contained 15 channels with 12 plants per channel for wild rocket and three floating trays for sea fennel. The postharvest experiment was submitted to mathematical analysis of data for each treatment: compost and peat in wild rocket and 'Peat leach', 'Compost leach', and 'Control' in sea fennel, package (OPP and PLA), and MAP storage duration (processing day and after 7 days) in both crops, and were subjected to multivariate analysis of variance (MANOVA). Means were compared by Tukey's test ($p = 0.05$) using Statgraphics Centurion (v.XIX, Stat Point Technologies, Inc., Warrenton, VA, USA).

### 3. Results and Discussion

#### 3.1. Head-Space Composition

The final atmosphere composition inside the packages was balanced between the product's respiration and the permeation of the gases through the packaging material (Table 1). The highest $CO_2$ concentrations were observed for rocket leaves grown in peat, and for sea fennel grown in both 'peat leach' and 'control' treatments, indicating a higher respiration rate for them compared to those cultivated in compost or in 'compost leach' treatment. Lower respiration rates can be associated with a longer shelf-life. However, for brief storage periods resembling those in short food supply chains, differences in shelf-life can be unseen. PLA bags allowed for obtaining an atmosphere with less $O_2$ and more $CO_2$ than OPP for both crops and, independently of the treatment, a better suitability of PLA for MAP was indicated since a higher modification related to air was reached. Coating one side of PLA with Kraft paper reduced the gas and water vapour exchange rate and allowed a more appropriate passive atmosphere modification by compensating for the higher gas permeability and water vapour transmission of PLA when compared to OPP. The effectiveness of MAP in prolonging rocket shelf-life has been widely studied. Wild rocket is usually minimally processed and packaged in plastic, with 5 to 10 kPa $CO_2$ and 5 to 10 kPa $O_2$ as suitable compositions for convenient shelf-life [30]. In general, wild and salad rocket species have a prolonged postharvest shelf-life when exposed to levels of $CO_2$ around 8–10 kPa, preserving sensory and microbiological quality as well as the content of health-promoting phytonutrients compared to air [31]. There are no relevant references in

the bibliography indicating the most appropriate MAP for sea fennel. However, previous reports indicated that after 6 days of storage at 5 °C, any significant modification in $O_2$ and $CO_2$ concentrations were observed and that slight atmosphere changes were detected after 12 days [26], thus confirming a rather low respiration rate for this crop.

**Table 1.** Atmosphere composition within OPP or PLA bags at the end of storage for 7 days at 4 °C of fresh-cut wild rocket and sea fennel cultivated in different substrates and their leachates, respectively.

| | $O_2$ (kPa) at Day 7 (*) | | $CO_2$ (kPa) at Day 7 (*) | |
|---|---|---|---|---|
| | **OPP** | **PLA** | **OPP** | **PLA** |
| Wild rocket | | | | |
| **Peat** | 14.7 ± 0.8 bA | 11.5 ± 2.2 aB | 6.2 ± 0.8 aB | 9.4 ± 2.2 aA |
| **Compost** | 16.8 ± 0.9 aA | 13.0 ± 1.4 aB | 4.1 ± 0.9 bB | 7.9 ± 1.4 aA |
| Sea fennel | | | | |
| **Peat leach** | 19.6 ± 0.3 aA | 16.9 ± 1.1 aB | 1.5 ± 0.3 abB | 3.8 ± 1.0 aA |
| **Compost leach** | 19.8 ± 0.3 aA | 15.1 ± 0.4 aB | 1.2 ± 0.2 bB | 5.4 ± 0.6 aA |
| **Control** | 19.1 ± 0.2 aA | 16.1 ± 0.9 aB | 1.9 ± 0.1 aB | 4.6 ± 0.6 aA |

Different lowercase letters within each column and species indicate significant differences among treatments, while different uppercase letters within each row indicate significant differences between packages at $p = 0.05$ according to Tukey's test. OPP: oriented polypropylene bags; PLA: polylactic acid bags. (*) At day 0, $O_2 = 21.02$ kPa and $CO_2 = 0.05$ kPa.

The atmospheres obtained in our experiments preserved the shelf-life of rocket and sea fennel leaves for 7 days at 4 °C. It is important to highlight the capacity of PLA for generating an atmosphere closer to that most appropriate for wild rocket storage and good for preserving the quality of sea fennel, indicating the suitability of this material as a substitute for those from typical plastic sources.

*3.2. Weight Loss*

Leafy vegetables are highly susceptible to water loss after harvest. Transpiration is the major cause of postharvest loss and poor quality, which causes wilt after about 3–5% water loss [32]. Values detected in our experiments were below that range. Correct packaging can prevent shriveling by ensuring high humidity inside the bag. Dehydration was higher for leaves stored in PLA than for those packaged in OPP, particularly for wild rocket, with those coming from compost and those that were OPP-packaged having the lowest values (Figure 1a). For sea fennel (Figure 1b) there were no differences between packages and growing media.

The water transmission rate for PLA was higher than for OPP (18 g m$^{-2}$ d$^{-1}$ vs. 330 g m$^{-2}$ d$^{-1}$, respectively) and, even though several properties of PLA-based packaging material have been found to be similar to petroleum-based films, the hygroscopic nature of PLA has been found to influence product quality [33]. The elevated water vapour transmission rate of PLA is related to its chemical composition. One of the main drawbacks of PLA for various applications is its sensitivity to hydrolysis in the presence of water, leading to a drastic decrease in molecular weight and degradation of mechanical properties [34]. Hydrolysis increases at elevated temperatures, especially above the glass transition temperature [35]. The water resistance of PLA composite can be significantly improved by the addition of microencapsulated polymethyl-methacrylate (PMMA), which in combination with PLA limits water diffusion [18]. PLA bags had 57.5% of their surface coated by Kraft paper, as an affordable industrial option to reduce water loss. More studies are needed for PLA blends and copolymers and for compounds that can be applied or added to improve the physical, mechanical, and barrier properties of PLA [36].

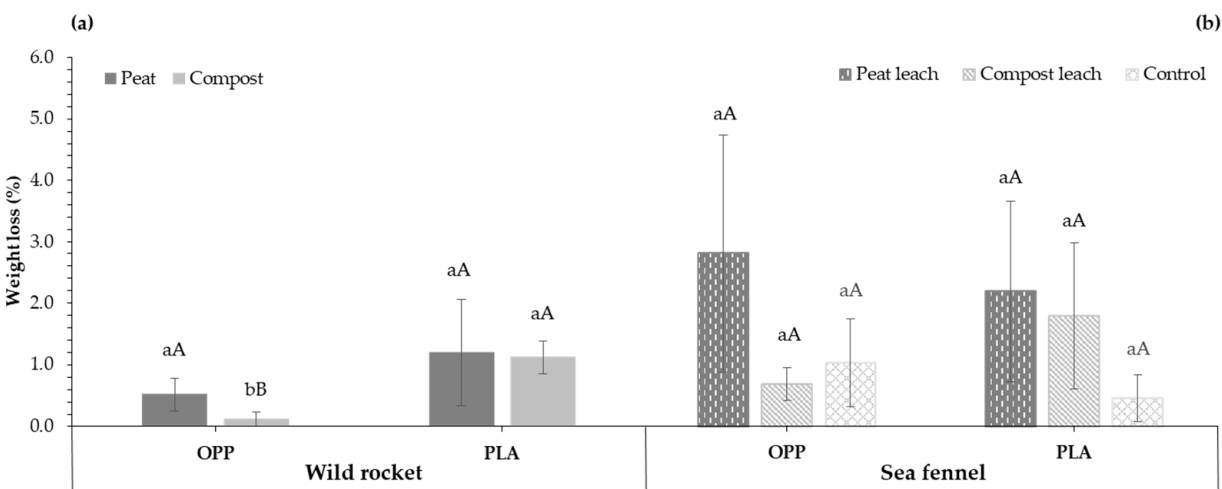

**Figure 1.** Weight loss of fresh-cut wild rocket (**a**) and sea fennel (**b**) cultivated in different growing media and their leachates, respectively, packaged in OPP or PLA bags for storage during 7 days at 4 °C. Different lowercase letters indicate significant differences among treatments for each vegetable, while different uppercase letters indicate significant differences between packages of each crop at $p = 0.05$ according to Tukey's test. OPP: oriented polypropylene bags; PLA: polylactic acid bags.

However, as will be discussed below, water loss measured for PLA-packaged wild rocket and sea fennel was not noticed by the trained panel; all the leaves showed a freshness aspect after 7 days at 4 °C. Independently of that, we hypothesise that water loss could have been significantly reduced by lowering the temperature during storage to reduce the vapour pressure deficit. Wilting is better prevented at 0 °C than at 4 °C or above, particularly if longer storage periods (>7 days) are intended. Moreover, breaks in the cold chain during distribution and retail must be avoided.

### 3.3. Microbial Quality

Before washing, microbial load was between 1.7–2.3 and 6.0–6.4 log CFU $g^{-1}$ for total mesophilic aerobic bacteria; between 0.9–1.5 and non-detected log CFU $g^{-1}$ for psychrophilic; and between 0.3–1.3 and 4.3–6.2 log CFU $g^{-1}$ for yeasts and moulds for wild rocket and sea fennel, respectively, while *Enterobacteriaceae* were not detected in any vegetable or in any treatment (data not shown) (Figure 2). Initial values of microbial counts for wild rocket before disinfection were low and typical for soilless cultivated vegetables when compared with standard cultivation [37] except for psychrophiles, and with lower values when compared with sea fennel. Washing with NaClO was highly effective for decreasing counts to undetectable levels in wild rocket. However, for sea fennel, the only decrease was detected for yeast and moulds when leaves were grown in 'peat leach'. The waxy surface of sea fennel can reduce disinfection efficacy and longer washing time or higher chlorine concentrations could be needed. Amoruso et al. [26] obtained similar values for microbial load at harvest in sea fennel grown on saline conditions in a floating system.

At the end of storage, the microbial load of wild rocket increased for all the treatments without significant differences between growing media or between packages. It is worth highlighting that the mesophilic counts observed in leaves for peat-cultivated plants that were packaged in PLA exhibited a lower increase compared to the other treatments. This may be because peat-grown, PLA-packaged rocket had the highest weight loss due to dehydration, resulting in an atmosphere less favourable for microbial growth.

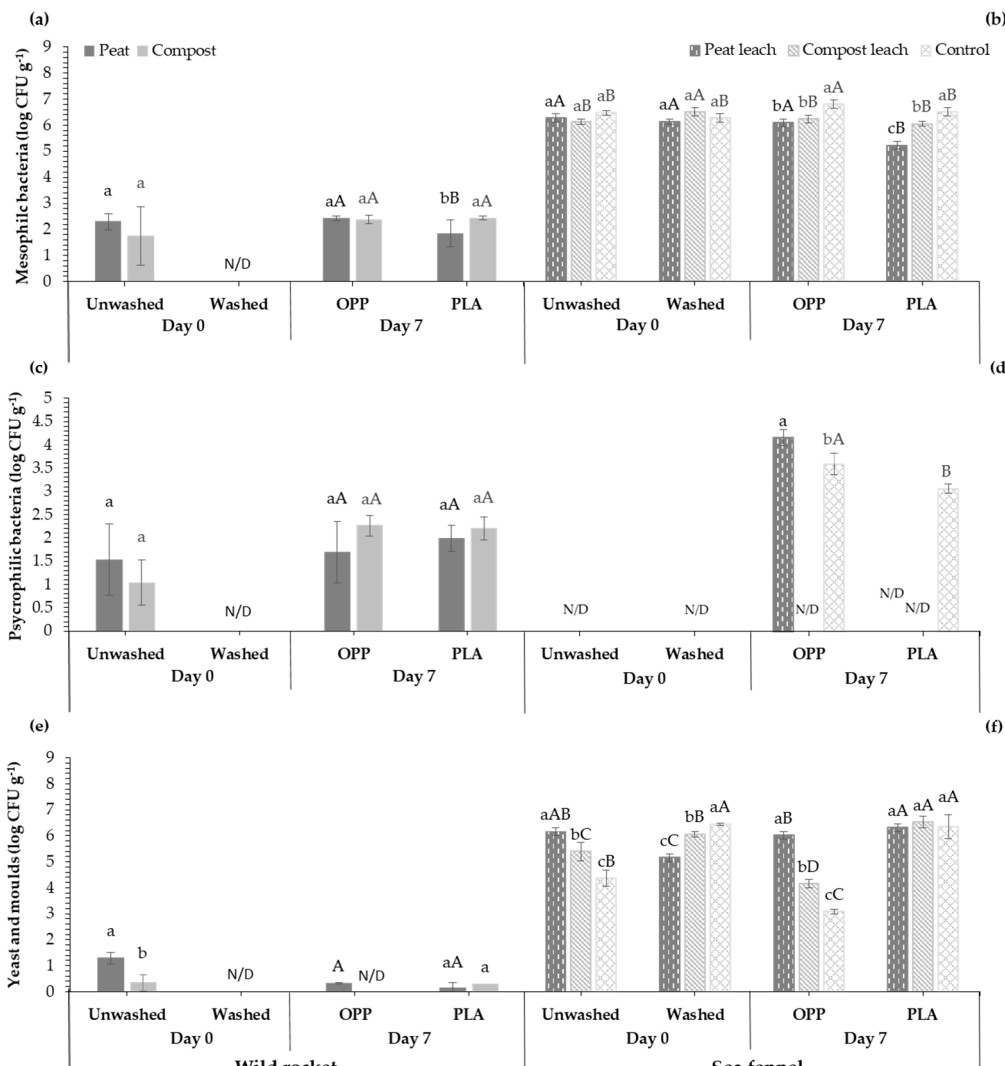

**Figure 2.** Mesophilic bacteria, psychrophilic bacteria, and yeast and mould counts (log CFU $g^{-1}$) of fresh-cut wild rocket (**a,c,e**) and sea fennel (**b,d,f**) cultivated in different growing media and their leachates, respectively, packaged in OPP or PLA bags, and stored during 7 days at 4 °C. Values at harvest (day 0) and at the end of storage (day 7). Different lowercase letters indicate significant differences among treatments for each vegetable, while different uppercase letters indicate significant differences between packages at $p = 0.05$ according to Tukey's test. N/D: non detected. OPP: oriented polypropylene bags; PLA: polylactic acid bags.

Results for sea fennel indicated a significant interaction among leachates, packaging, and storage for mesophilic and psychrophilic bacteria and yeasts and moulds. Mesophilic counts of sea fennel slightly changed during storage, with the leaves of plants grown in peat leachates stored in PLA showing the lowest values. PLA was also more effective for reducing psychrophiles growth, especially for peat and compost drainages, while yeast and mould were grown at a lower rate in OPP than in PLA. Modified atmosphere packaging is commonly used for decreasing microbial growth on perishable commodities. The atmosphere reached in our experiments for sea fennel was adequate because anaerobic microorganisms were not detected. However, since microbial growth was poorly inhibited, higher $CO_2$ concentrations could be needed for reaching a bacteriostatic effect. For better control of the microbial safety of fresh-cut sea fennel, the packaging design can be improved. Using different composites to control microbial growth and other aspects of produce metabolism is a critical area of research that can have significant practical applications for the food industry. By combining different composites and techniques, it may be possible to

enhance the shelf-life of food products as has been demonstrated by Singh et al., 2023 [23], in recent research.

It is known that values over 7 log CFU g$^{-1}$ for mesophilic and psychrophilic bacteria can be associated with a shorter shelf-life of fresh-cut vegetables. In our experiments, and independently of the treatments and of the crop, values above those mentioned were not found in any case, as reported by Giménez et al. [38].

### 3.4. Colour

Non-remarkable differences in colour were observed between leaves of wild rocket cultivated in peat or compost at harvest (Table 2). Additionally, PLA and OPP bags were favourable for keeping colour during storage with any relevant change for any of the colour parameters analysed. It could be related to the atmosphere modification surrounding the plant product that would be appropriate for avoiding the loss of green colour. For sea fennel, the colour was darker when obtained from the 'control' and from 'compost leach' than from 'peat leach'. During storage, only an increase in L* values was detected for all the sea fennel treatments indicating a slight increase in lightness, probably due to the presence of salt crystals on leaves as reported by Amoruso et al. [26]. Likewise, a slight trend towards an increase in L* was observed after 7 days of storage for wild rocket leaves that were grown in compost.

**Table 2.** Colour parameters for fresh-cut wild rocket and sea fennel cultivated in different growing media and their leachates, respectively, packaged in OPP or PLA bags. Day 0 = at harvest. Day 7 = after storage during 7 days at 4 °C.

| | L* | | | Hue | | |
|---|---|---|---|---|---|---|
| | Day 0 | Day 7 | | Day 0 | Day 7 | |
| | | OPP | PLA | | OPP | PLA |
| **Wild rocket** | | | | | | |
| **Peat** | 45.3 ± 4.6 aA | 48.3 ± 5.1 aA | 47.2 ± 4.3 aA | 122.7 ± 1.3 aA | 122.3 ± 1.6 aA | 122.3 ± 2.0 bA |
| **Compost** | 44.7 ± 2.1 aB | 48.2 ± 3.7 aA | 47.7 ± 4.9aA | 122.4 ± 0.8 aAB | 121.9 ± 1.7 aB | 123.3 ± 1.7 aA |
| **Sea fennel** | | | | | | |
| **Peat leach** | 42.7 ± 4.2 aB | 46.9 ± 3.1 aA | 46.9 ± 3.2 aA | 115.0 ± 4.4 bA | 115.8 ± 4.1 cA | 115.5 ± 4.5 bA |
| **Compost leach** | 38.1 ± 4.1 bB | 44.2 ± 4.9 aA | 42.9 ± 3.5 bA | 118.3 ± 9.1 abA | 119.3 ± 2.1 bA | 123.1 ± 3.1 aA |
| **Control** | 33.4 ± 4.4 cB | 39.2 ± 1.9 bA | 39.9 ± 4.6 dA | 122.8 ± 2.5 aA | 123.5 ± 4.3 aA | 125.4 ± 3.5 aA |

Different lowercase letters within each column indicate significant differences among treatments for each vegetable on days 0 and 7, while different uppercase letters within each row indicate significant differences between packages at $p$ = 0.05 according to Tukey's test. OPP: oriented polypropylene bags; PLA: polylactic acid bags.

The absence of anaerobic condition, which might lead to acidic degradation of chlorophyll and, lastly, to senescence and loss of greenness, avoided relevant colour alterations to the leaves in the products.

Reductions in the hue angle were not drastic. On the contrary, it would indicate a change from green to yellow. Green colour is an important quality parameter of leafy vegetables at the time of purchase as it indicates the freshness of the product [39].

### 3.5. Sensory Evaluation

Sensory quality at harvest, when colour, aroma, texture, and dehydration were considered, was similar for rocket coming from both growing media (Figure 3a). However, wild rocket grown in peat had a worse overall quality due to a lower flavour punctuation. That aspect was related to an exceptionally strong spicy flavour that was criticised by the panel members as reflected in the tasting notes (data not shown). For sea fennel the overall sensory quality decreased in leaves from plants treated with peat leachate respective to the control and the compost leachate (Figure 3d). After 7 days of storage, rocket quality in biodegradable and plastic packages remained as good and acceptable for consumption, independently of the growing media (Figure 3b,c). In agreement with the data obtained

from the colourimetric measurements, yellowness was not detected in any treatment. The slight loss in texture for PLA-stored wild rocket seemed to have a trend similar to that for dehydration, with treatments from biodegradable bags showing the poorer texture. Sea fennel grown in peat leachate showed a worse overall quality after 7 days of storage in both biodegradable and plastic packages than that from the other treatments (Figure 3e).

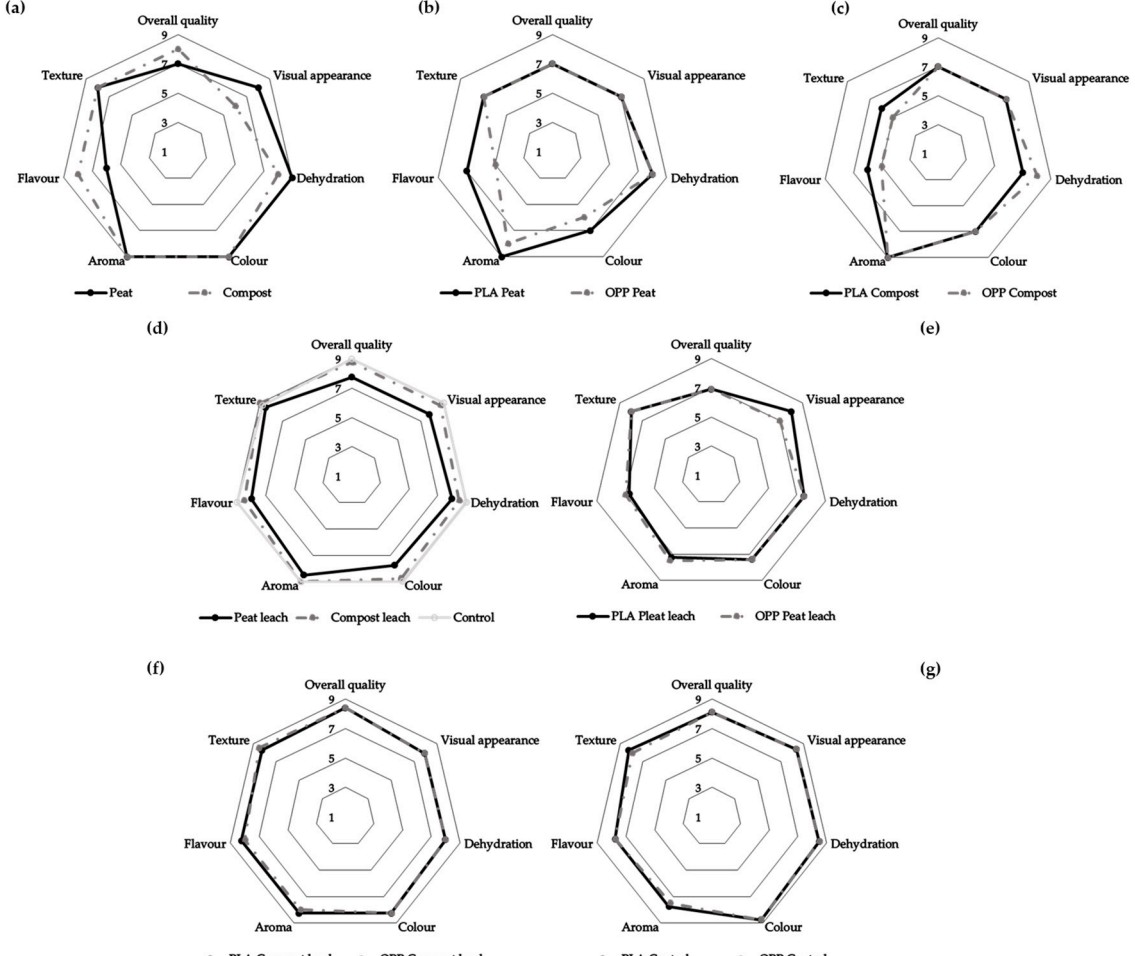

**Figure 3.** Sensory quality of fresh-cut wild rocket (**a–c**) and sea fennel (**d–g**) cultivated in different substrates and their leachates, respectively, packaged in OPP or PLA bags and stored during 7 days at 4 °C. (**a**) Rocket leaves from plants grown in peat and compost, at harvest. (**b**) Rocket leaves from plants grown in peat after 7 days in OPP or PLA bags. (**c**) Rocket leaves from plants grown in compost after 7 days in OPP or PLA bags. (**d**) Sea fennel leaves from plants treated with 'peat leach', 'compost leach', and 'control', at harvest. (**e**) Sea fennel leaves from plants treated with 'peat leach' after 7 days in OPP or PLA bags. (**f**) Sea fennel leaves from plants treated with 'compost leach' after 7 days in OPP or PLA bags. (**g**) Sea fennel leaves from 'control' plants after 7 days in OPP or PLA bags. OPP: oriented polypropylene bags; PLA: polylactic acid bags.

The packaging of green leafy vegetables can postpone senescence and yellowing, but a drawback is the risk of anaerobic respiration preceding the development of an olive-brown colour. MAP not correctly designed can lead either to an internal atmosphere composition close to atmospheric air at too high transmission rates or to anaerobic respiration due to too low transmission rates of gases through the package. This would lead to the onset of symptoms of tissue degradation and appearance of off-odours [40]. In contrast, atmospheric air leads to senescence and loss of green colour [38]. The MAP obtained in our experiments was appropriate for keeping flavour in both PLA and OPP bags. The texture is another important quality parameter of green leafy vegetables [41]. The senescence of vegetables is

a degradation process, where the cell walls are broken down. Cell collapse may also be induced by too low $O_2$ in the intracellular spaces of the living tissues [42], resulting in loss of texture. In our experiments, only a moderated loss of texture was observed for rocket leaves obtained from compost and packaged with PLA, probably related to the higher water loss observed in them when compared to those grown in the same growing medium and packaged with OPP.

Finally, the overall quality, which determined the degree of acceptance, was also influenced by flavour. Guijarro-Real et al. [43] reported that acceptance was mainly related to taste and pungency, with rocket (*Diplotaxis tenuifolia* (L.) in our case) being well accepted only by a cohort of consumers that enjoy spicy flavours. On the other hand, sensory changes in sea fennel were not drastic, indicating that a longer shelf-life could be obtained than that of wild rocket. Sea fennel likely would have the longest postharvest life in terms of sensory quality, which may be attributed to its lower respiration rate compared to wild rocket, as well as to the natural waxy coating on its leaves that reduces moisture loss by retaining it within the plant tissue.

*3.6. Nitrate Content*

Nitrates are typically accumulated in some green leafy vegetables such as lettuce, spinach, and rocket [44]. Particularly, rocket has been classified as a plant with typically high concentrations of $NO_3^-$ (>2500 mg kg$^{-1}$ FW, [45]) with wild rocket having twice the concentration of cultivated rocket (*Eruca vesicaria*) [46]. The maximum levels set for nitrates according to the European Commission [47] for rocket are 6000 and 7000 mg $NO_3^-$ kg$^{-1}$ FW, if harvested from 1 April to 30 September and from 1 October to 31 March, respectively. However, specific limits for sea fennel are not legally stated yet.

Significant differences in nitrate concentration were found between wild rocket leaves harvested from plants grown in different growing media, with those cultivated in compost having the highest concentrations (Figure 4a). This is not surprising, as the content of total N in compost was four-fold that in peat [25]. The nitrogen in the compost was hence (at least partly) released into the drainage nutrient solution which had, indeed, a nitrate concentration three to four times higher than the drainage obtained from peat (data not shown). This effect, observed for the main crop, was not reflected in the nitrate content in sea fennel, where peat and compost leachates were used (Figure 4b), without any difference between them. However, the nitrate content of sea fennel leaves obtained from the control treatment had higher concentrations. These concentrations were higher than those obtained by other authors [15,26].

Differences were maintained during storage, even when a slightly decreasing trend was observed with time (Figure 4) for both the main and the secondary crop. The type of package did not have any influence on nitrate metabolism during shelf-life. It is known that nitrate might have both health concerns, such as methaemoglobinaemia and cancer, and health benefits, such as positive cardiovascular effects and an increase in human defense against gastroenteritis [44,45]. Although the scientific community has not yet reached a consensus on the issue, the values obtained from the experiments were relatively high. Our findings suggest that this may be attributed to the tendency of wild rocket leaves to accumulate nitrates more efficiently than other leafy vegetables. Wild rocket's short biological cycles, high rate and extent of N recovery, and its capacity to accumulate nitrates all indicate that it could function as a hyperaccumulator of nitrates. However, it is important to note that these values remained below the legally established limits for commercialisation [48]. On the other hand, sea fennel did not accumulate as much nitrate content as wild rocket leaves (Figure 4).

Some strategies can be used to reduce the nitrate content, particularly in hydroponic systems, to avoid its accumulation in rocket leaves, which would hamper the rocket being placed on the market [46,49].

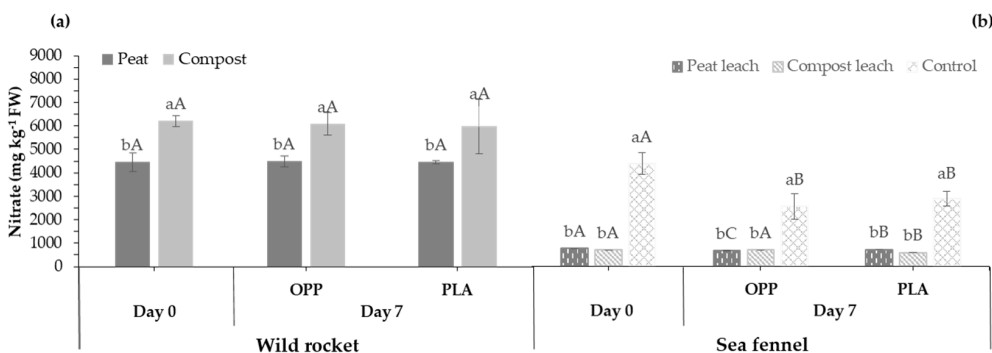

**Figure 4.** Nitrate content of fresh-cut wild rocket (**a**) and sea fennel (**b**) cultivated in different growing media and their leachates, respectively, packaged in OPP or PLA bags and stored during 7 days at 4 °C. Values at harvest (day 0) and at the end of storage (day 7). Different lowercase letters indicate significant differences among treatments, while different uppercase letters indicate significant differences between packages at $p = 0.05$ according to Tukey's test. OPP: oriented polypropylene bags; PLA: polylactic acid bags.

### 3.7. Vitamin C

The vitamin C content measured as ascorbic acid (AA) and dehydroascorbic acid (DHA) ranged between 63 and 101 mg kg$^{-1}$ FW for wild rocket (Figure 5a). The highest content of vitamin C at harvest, as well as after storage, was observed for rocket grown in compost. Previous reports indicated that the vitamin C of rocket salad can be positively affected when organic amendments are added [50]. Values found in this study are lower than those previously reported for rocket salad cultivated in soil [51] but similar to those of Duyar and Kiliç [52] for plants grown in a floating system. The vitamin C content in leaves can be influenced by growing conditions, particularly by the $NO_3-/NH_4^+$ ratio in the nutrient solution [53]. Observations revealed no significant vitamin C degradation after 7 days, which suggests that both the low temperature and the MAP were effective at preventing its degradation. Bonasia et al. [54] stated that vitamin C degradation for wild rocket could occur after 7 days at 5 °C, but would be lower than 21% and would depend on the initial values. Differences due to the package were not detected, indicating that PLA was as suitable as OPP for keeping the initial values of vitamin C during cold storage.

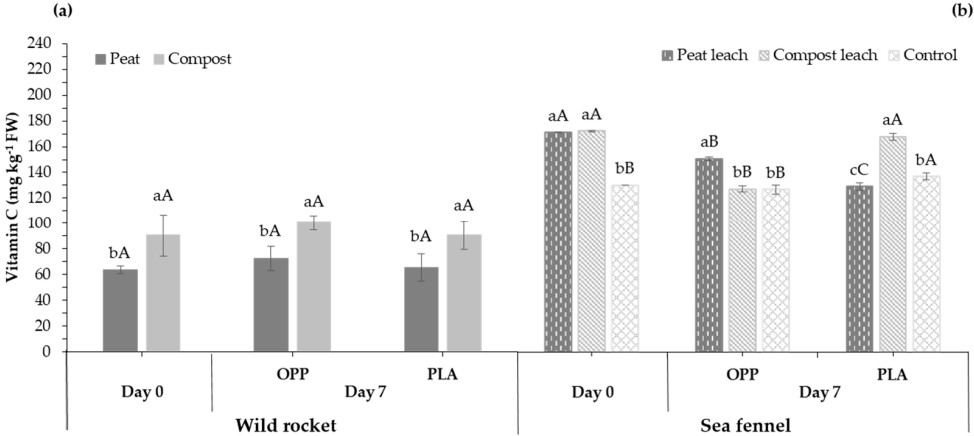

**Figure 5.** Vitamin C content of fresh-cut wild rocket (**a**) and sea fennel (**b**) cultivated in different growing media and their leachates, respectively, packaged in OPP or PLA bags and stored during 7 days at 4 °C. Values at harvest (day 0) and at the end of storage (day 7). Different lowercase letters indicate significant differences among treatments for each vegetable, while different uppercase letters indicate significant differences between packages at $p = 0.05$ according to Tukey's test. OPP: oriented polypropylene bags; PLA: polylactic acid bags.

The vitamin C content for sea fennel was higher than in wild rocket (Figure 5b). Interestingly, compost and peat leachates significantly increased the vitamin C in leaves compared with the control. A slight decrease was observed during storage, with the highest content in the plants grown in compost leach and stored in PLA. That decrease could be avoided by a MAP with lower oxygen concentrations resembling the atmosphere composition observed for wild rocket. These results confirm that sea fennel is a vegetable rich in vitamin C, quite stable during storage in a biodegradable package, and in agreement with those results previously reported by Renna [55].

### 3.8. Total Phenolics Content and Total Flavonoids Content

Results obtained for total phenolics content (Figure 6a,c) indicate that the initial values were similar to those previously reported by Gutiérrez et al. [56] for wild rocket and considerably higher than those reported by Amoruso et al. [26] for sea fennel.

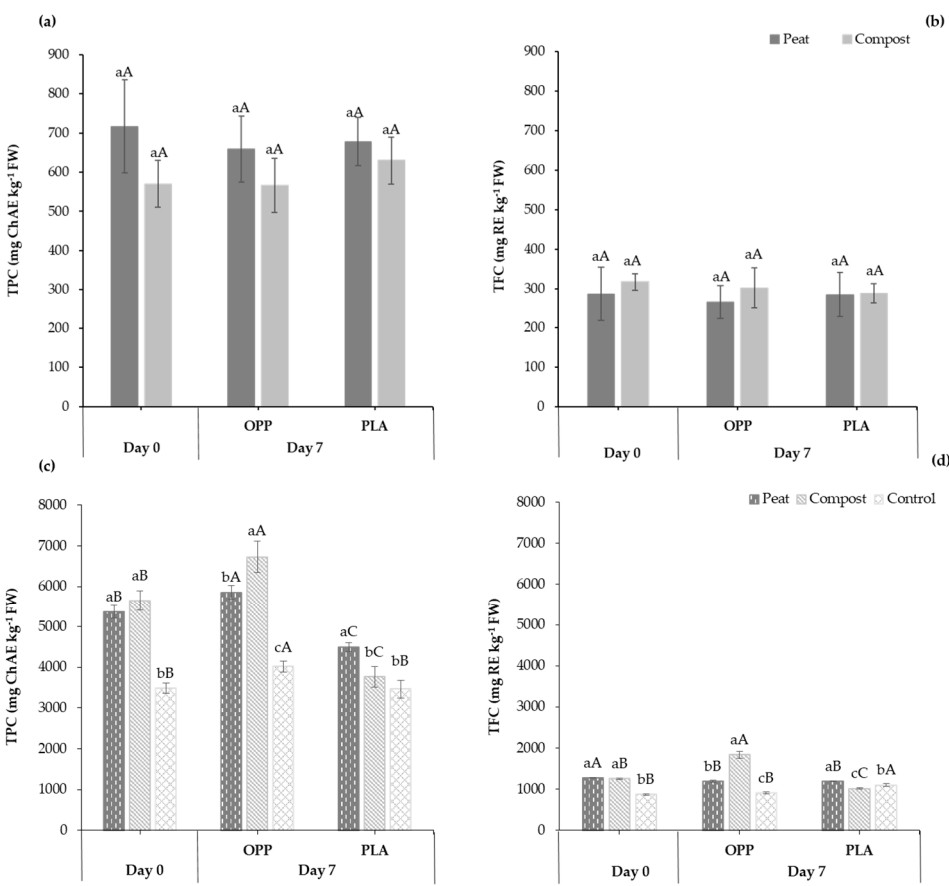

**Figure 6.** Total phenols (**a**–**c**) and total flavonoids (**b**–**d**) of fresh-cut wild rocket (**a**,**b**) and sea fennel (**c**,**d**), respectively, cultivated in different growing media and their leachates, respectively, packaged in OPP or PLA bags and stored during 7 days at 4 °C. Values at harvest (day 0) and at the end of storage (day 7). Different lowercase letters indicate significant differences among treatments for each vegetable, while different uppercase letters indicate significant differences between packages at *p* = 0.05 according to Tukey's test. OPP: oriented polypropylene bags; PLA: polylactic acid bags.

Any remarkable trend was observed after 7 days of storage for wild rocket. Most of the changes in TPC in leafy vegetables have been reported to be detected after longer storage times and/or higher temperatures than those assessed here [40,56]. In general, the leaves from rocket grown in peat showed a trend of higher values of TPC but still non-significant. For sea fennel, the highest values were observed for plant leaves grown in peat and compost leachates, with the lowest being found in the control. Then, they were quite stable during storage for the OPP bags, with a decrease for the PLA. However,

values were considerably high. Phenolic compounds play a key role in the protection of plant tissues against abiotic stress [56]. Growing plants with leachates of peat and compost would trigger phenolics accumulation.

Similarly, flavonoid concentrations (Figures 6b and 6d, respectively) were similar to those reported by Bell and Wagstaff [12] and Amoruso et al. [26] for wild rocket leaves and sea fennel, respectively. As a part of phenolics compounds, flavonoids were shown to represent more than 50% of them for all the treatments. The storage did not have any influence on flavonoid values with ranges practically constant for all the treatments. Flavanol derivatives of quercetin and kaempferol had been detected in wild rocket with quercetin derivatives being the main compounds in wild rocket, while kaempferol derivatives were the main compounds in salad rocket [51]. For a better understanding of flavonoid pathways, it would be advisable to analyse the flavanols profile, especially for sea fennel, where scarce information is still available. The results indicated that sea fennel is a vegetable rich in flavonoids and that the leachates, especially those based on compost, have a positive influence on flavonoid accumulation.

### 3.9. Total Antioxidant Capacity

At harvest, the antioxidant capacity in rocket leaves was similar in all the treatments and within the ranges previously reported by other authors [56] and particularly for this species [51], varying from 1.450 to 1.680 mg TE kg$^{-1}$ FW (Figure 7a). As observed for TPC, there were no significant variations in TAC during storage at 4 °C. Furthermore, there were no significant differences between rocket plants cultivated in peat or compost. The same happened for the type of package which did not affect the total antioxidant capacity of the leaves. For short storage periods at low temperatures (<5 °C), TAC has been shown to be constant indicating a high stability of rocket in its antioxidant system. That stability has been previously reported even when the leaves had to cope with strong physical stresses [56]. Sea fennel (Figure 7b) had higher TAC content when plants were cultivated with peat and compost leachates (4.672 and 3.643 mg TE kg$^{-1}$ FW, respectively) when compared to the control (971 mg TE kg$^{-1}$ FW), indicating that antioxidants can be favoured with sea fennel as a secondary crop and wild rocket as the primary. To our knowledge, this is the first time that this beneficial effect of leachates on nutritional quality has been identified. Stressing growing conditions, such as the accumulation of toxic ions in the leaves (i.e., Cl$^{-}$, Na$^{+}$ 388) and/or the alkaline pH of the NS, can affect sea fennel plants and enhance their phytochemical content [26]. It is well stablished that plants cope with abiotic stress by altering metabolic processes producing reactive oxygen species and stimulating antioxidant activity to scavenge free radicals and ion chelators [56]. TAC decreased during storage, with the highest values for the OPP bags.

For wild rocket, TAC was related to the content of flavonoids and polyphenols showing the same trend, that is, a lack of relevant changes during storage. Previous studies of the antiradical activity in wild rocket leaves have shown that it is correlated to polyphenol content and flavonoids, as well as to vitamin C, these being the major antioxidants of Brassica vegetables [57]. However, in addition to them, other constituents could exhibit antioxidant properties such as vitamin E and carotenoids for wild rocket [51] and sea fennel [26]. In the case of sea fennel, antioxidants such as ascorbic acid, phenolic compounds, and, particularly, flavonoids, as previously indicated, can contribute to these high values.

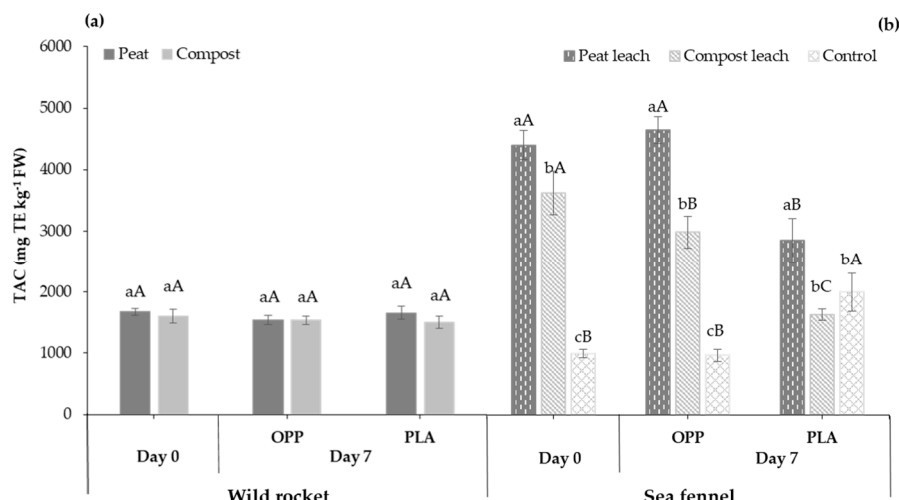

**Figure 7.** Total antioxidant capacity of fresh-cut wild rocket (**a**) and sea fennel (**b**) cultivated in different growing media and their leachates, respectively. Packaged in OPP or PLA bags and stored during 7 days at 4 °C. Values at harvest (day 0) and at the end of storage (day 7). Different lowercase letters indicate significant differences among treatments for each vegetable, while different uppercase letters indicate significant differences between packages at $p = 0.05$ according to Tukey´s test. OPP: oriented polypropylene bags; PLA: polylactic acid bags.

### 4. Conclusions

The present study lays the groundwork for future research into evaluating biodegradable packages for the short food supply chain of leafy vegetables grown in a cascade cropping system. Results of this research showed that packaging fresh-cut wild rocket and sea fennel with a PLA-based film is a feasible alternative to common plastic used in the fresh-cut industry, avoiding the waste of polymers in landfills. The biodegradable film has good oxygen barrier properties for packaging wild rocket. For sea fennel, a lower gas exchanging area or a reduced head-space can be recommended to obtain a MAP with higher $CO_2$ and lower $O_2$ than observed here. To our knowledge, this is the first report in evaluating a compostable package for the short food supply chain of leafy vegetables. A shelf-life of 7 days at 4 °C, appropriate for the purpose of the ready-to-eat-sector, can be achieved without any relevant detrimental change in quality or microbial safety. Moreover, soilless cultivation with compost as growing media allowed for obtaining wild rocket with lower water loss and respiratory activity when compared to that cultivated in peat. At the same time, sea fennel, as a secondary crop grown using the leachates derived from the wild rocket growing medium, showed a higher nutritional quality than that observed for sea fennel cultivated with standard nutrient solution. The combination of the cascade cropping system using compost as growing medium or its leachate and PLA for packaging was demonstrated to be feasible for wild rocket and sea fennel production, opening a wide range of possibilities for a more environmentally friendly production and commercialisation of fresh-cut vegetables. Our study focused on a specific biodegradable packaging material that yielded satisfactory results. However, there is potential to further improve the shelf-life of food products, reduce food waste, and enhance food safety by exploring different composite materials and production techniques for biodegradable packaging. As the use of less plastic for packaging becomes increasingly important in sustainable agriculture, future research should prioritise the development of eco-friendly alternatives. We hope that our study will contribute to this important field and inspire further investigation into effective methods for food preservation.

**Author Contributions:** Conceptualisation, J.A.F., P.A.G., C.E.-G. and J.O.; methodology, A.G., F.A., R.R.B., A.S. and V.M.G.-C.; investigation, A.G., F.A., P.A.G., V.M.G.-C. and R.R.B.; resources, J.A.F., C.E-G. and J.O.; data curation, V.M.G.-C., A.G. and C.E.-G.; writing—original draft preparation, P.A.G., C.E.-G., A.G. and J.A.F.; writing—review and editing, P.A.G., C.E.-G., A.G., J.A.F. and A.S.;

supervision, J.A.F., J.O., P.A.G. and C.E.-G.; funding acquisition, J.A.F. and J.O. All authors have read and agreed to the published version of the manuscript.

**Funding:** The research was supported by Grant PID2020-114410RB-I00 funded by MCIN/AEI/ 10.13039/501100011033, and Grant AGROALNEX funded by Comunidad Autónoma de la Región de Murcia through Fundación Séneca—Agencia de Ciencia y Tecnología de la Región de Murcia and European Union NextGenerationEU. Angelo Signore was also funded by a grant of the Re-qualification of the Spanish University System, Maria Zambrano modality (Grant UP2021-033 funded by Ministerio de Universidades and by the "European Union NextGenerationEU/PRTR"). Víctor Gallegos was also funded by a grant of the Re-qualification of the Spanish University System, Margarita Salas modality, by the University of Almería (Grant UP2021-004 funded by Ministerio de Universidades and by the "European Union NextGenerationEU/PRTR"). Almudena Giménez was also funded by a grant of the Re-qualification of the Spanish University System, Margarita Salas modality, by the University of Cartagena (Grant UP2021-004 funded by Ministerio de Universidades and by the "European Union NextGenerationEU/PRTR").

**Data Availability Statement:** Not applicable.

**Acknowledgments:** Thanks are also due to Noelia Durán López for technical assistance at the laboratory.

**Conflicts of Interest:** The authors declare no conflict of interest.

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
