# Peer review of "Biodegradable Food Packaging of Wild Rocket (Diplotaxis tenuifolia L. [DC.]) and Sea Fennel (Crithmum maritimum L.) Grown in a Cascade Cropping System for Short Food Supply Chain"

_horticulturae, doi:10.3390/horticulturae9060621_

Round 1

Reviewer 1 Report

The manuscript entitled Biodegradable food packaging of wild rocket (Diplotaxis tenuifolia L. [DC.]) and sea fennel (Crithmum maritumum L.) grown in a cascade cropping system for short food supply chain shows results related to the cultivation by CCS and the storage by two methods of wild rocket and sea fennel. The manuscript is well-written and the topic is sound. However, there are issues that authors must attend to. Below are the comments.

-Section 2.1. What were the cultivation conditions for the wild rocket? 

-Line 131. What was the rationale for rinsing the plant material with tap water? Could the procedure contaminate the material?

-Lines 180-181. What was the rationale for analyzing by duplicate each of the triplicates?

-Line 219. Why the samples were derivatized? Since the detection at HPLC was by a photodiode array detector. 

-Line 243. Define TPTZ.

-Table 1. Was it carried out Tukey test for comparing pairs of packages? The authors must carry out a student T-test when analyzing pairs of treatments.

-Figure 1. How do the authors explain such big standard deviations?

-Figures 5-7. It is not clear which data belongs to day 0 and to day 7.

-

The MS was written in good English.

Author Response

The manuscript entitled Biodegradable food packaging of wild rocket (Diplotaxis tenuifolia L. [DC.]) and sea fennel (Crithmum maritimum L.) grown in a cascade cropping system for short food supply chain shows results related to the cultivation by CCS and the storage by two methods of wild rocket and sea fennel. The manuscript is well-written and the topic is sound. However, there are issues that authors must attend to.

Thank you very much for your immediate answer. We agree with all the points raised by you. We have carefully reviewed your suggestions and incorporated them into the latest version of the manuscript. The specific changes made are listed below:

-Section 2.1. What were the cultivation conditions for the wild rocket? 

A more detailed description of cultivation conditions for wild rocket has been added in the revised version of the manuscript.

-Line 131. What was the rationale for rinsing the plant material with tap water? Could the procedure contaminate the material?

Thank you for pointing that out. After disinfecting the product with chlorinated water (150 ppm), it is essential to rinse it thoroughly to ensure that the final chlorine residue is below 5 ppm, in compliance with legal limits for fresh salads in EU countries where chlorine is allowed as disinfectant. This can be achieved by immersing the product in cold, potable tap water (4°C) for at least 1 min. This process effectively removes any residual chlorine from the surface of the produce, making it safe for consumption. Since the product is rinsed with potable tap water, the likelihood of the material being contaminated is rare.

-Lines 180-181. What was the rationale for analyzing by duplicate each of the triplicates?

We duplicated each sample to minimize the impact of contamination and ensure reliable results during microbial analysis. This helps to overcome any potential issues that may arise from contamination or variability between measurements. By duplicating the samples, we can detect any inconsistencies or outliers in the results and take appropriate steps to address them. This approach enhanced the accuracy of our findings, allowing us to draw more reliable conclusions from the data.

-Line 219. Why the samples were derivatized? Since the detection at HPLC was by a photodiode array detector. 

To detect dehydroascorbic acid (DHA), the oxidized form of ascorbic acid (vitamin C), we needed to perform derivatization. Specifically, DHA reacts with 1,2-o-phenylenediamine dihydrochloride (OPDA) to form a fluorescent condensation product that can be detected by the photodiode array detector at 348 nm. In our analysis, we expressed the amount of vitamin C as the sum of DHA and AA (ascorbic acid) to account for both the oxidized and reduced forms of the vitamin. This approach enables us to assess the overall vitamin C content of the sample and ensure more reliable results. We have included additional details in the revised version of the manuscript.

-Line 243. Define TPTZ.

Sorry by the omission. We have added the meaning of TPTZ (2,4,6-tripyridyl-S-triazine), an analytical reagent for the spectrophotometric determination of antioxidants, in the revised version.

-Table 1. Was it carried out Tukey test for comparing pairs of packages? The authors must carry out a student T-test when analyzing pairs of treatments.

Thank you for asking for clarification. While a t-test may be suitable for comparing pairs of treatments, such as the different kinds of packaging materials (PLA and OPP), our study involved comparisons between multiple treatments, including growing media and different combinations of treatments. Therefore, we considered performing a multivariate analysis using Tukey's test, which is a more robust and consistent method. By performing both row and column comparisons, we were able to systematically analyze the data and draw conclusions about the effects of different treatments on atmosphere composition.

-Figure 1. How do the authors explain such big standard deviations?

After reviewing the data, we have identified a minor mistake and have now corrected the values of standard deviations to accurately reflect the variability in our samples and replace the figure for the correct one. However, we acknowledge that the standard deviations are still high. We believe that this variability may be due to several factors, including the inherent differences between samples, as well as potential overlaps between packages during storage. Packages that were directly exposed to the cold room conditions tended to lose more weight than those on the lower rows covered by others. Despite these variations, however, our results showed that the average weight losses were below the limit values from which wilting can be detected (3-5%).

-Figures 5-7. It is not clear which data belongs to day 0 and to day 7.

Thank you again for the comment. We have reviewed the figures 5 to 7 and made appropriate adjustments to better differentiate the data from day 0 from those of day 7. We trust that these changes will improve the clarity and make it easier for readers to interpret the data. Improved figures are now included in the revised version of the manuscript.

Reviewer 2 Report

The manuscript entitled Biodegradable food packaging of wild rocket (Diplotaxis tenui folia L. [DC.]) and sea fennel (Crithmum maritumum L.) grown in a cascade cropping system for short food supply chain is well drafted. Though title indicating degradable term and degradable bags have been used in the study, to prove the efficacy of the experiment degradability test for packaging bag is suggested. Moreover, a few suggestions require to improve the manuscript visibility among the researchers.

1. Suggested to include other recent technology in improvement of shelf-life of post harvest vegetable/fruit (example: https://doi.org/10.1007/s10924-022-02596-x) in introduction. Also elaborate the advantage of techniques used for shelf-life extension.

2. Suggested to add more discussion on effects on microbial growth, weight loss, and headspace composition due to leaching or interaction of packaging components with the wild rocket and sea fennel.

3. Suggested to report the calculated browning index based on color analysis

4. Suggested to add-on the results of antioxidant efficacy of other radical scavengers and compared the results.

5. Minor grammatical errors are presented/highlighted in the attached pdf file, please correct it.

Thanks and good luck

Minor editing of English language required

Author Response

The manuscript entitled Biodegradable food packaging of wild rocket (Diplotaxis tenui folia L. [DC.]) and sea fennel (Crithmum maritimum L.) grown in a cascade cropping system for short food supply chain is well drafted. Though title indicating degradable term and degradable bags have been used in the study, to prove the efficacy of the experiment degradability test for packaging bag is suggested. Moreover, a few suggestions require to improve the manuscript visibility among the researchers.

Thank you for bringing this to our attention and helping us to improve the quality of our work.

  1. Suggested to include other recent technology in improvement of shelf-life of post harvest vegetable/fruit (example: https://doi.org/10.1007/s10924-022-02596-x) in introduction. Also elaborate the advantage of techniques used for shelf-life extension.

Thank you for your feedback. We appreciate your input and have taken your suggestion into consideration. We have updated the Introduction section to include recent advances related to composites, in accordance with the scope of the manuscript. We have also highlighted the importance of extending the shelf life of food products with these techniques.

  1. Suggested to add more discussion on effects on microbial growth, weight loss, and headspace composition due to leaching or interaction of packaging components with the wild rocket and sea fennel.

As noted in Section 2.3 of the manuscript, global migration analysis was performed to evaluate the potential interaction between the packaging components and the food products. Specifically, we used modified polyphenylene oxide (MPPO) as a solid food simulant for the food contact side and tested the samples. The results showed that the global migration was below the limit of quantification (as per DIN EN 1186:2002-07/2002-12), indicating that the leaching of packaging components was negligible. Furthermore, the storage time in our study was intentionally short, as we focused on a short food supply chain. As a result, there was a minimal time elapsed between packaging and the end of the shelf life, which limited the possibility of observing any significant interaction between the packaging and the food products. We hope that this information clarifies our methodology and provides a better understanding of our findings.

Thanks to your observation, we have added a paragraph in the discussion (Section 3.3) to highlight the importance of the use of different composites that can help on the control of microbial growth and other aspects of produce metabolism by the interaction of packaging components with the fresh produce.

  1. Suggested to report the calculated browning index based on color analysis

Thank you for the comment. Actually, we calculated browning index based on two different equations: Kasim, R. and Kasim M.U. (http://dx.doi.org/10.1590/1678-457X.6523, 2015) and on Ruangchakpet, A. and Sajjaanantakul, T. Nat. Sci., 41. (2007). However, the obtained values were not significant, as there was no visible browning in the samples. Browning index has been typically used for fresh-cut products that have undergone mechanical processing such as shredding, slicing, or cutting. Our samples were not subjected to such processing, so the index was not an appropriate measure for our study.

  1. Suggested to add-on the results of antioxidant efficacy of other radical scavengers and compared the results.

The efficacy of radical scavenging was assessed using three different assays, FRAP, DPPH, and ABTS+. Among these, the FRAP assay yielded the most consistent results, especially for sea fennel. Given that sea fennel is a succulent plant, it is likely that the FRAP assay was more sensitive in detecting its antioxidants. However, further research is required to elucidate the underlying mechanisms of each method for Crithmum maritimum. By expressing the data using the FRAP method, it was more accurate to compare the values obtained for wild rocket and sea fennel.

  1. Minor grammatical errors are presented/highlighted in the attached pdf file, please correct it.

In the new version of the manuscript, all minor grammatical errors have been corrected. We appreciate your careful attention to detail and thank you for helping to improve the quality of the manuscript.

Reviewer 3 Report

The authors of the paper "Biodegradable food packaging of wild rocket (Diplotaxis tenui-folia L. [DC.]) and sea fennel (Crithmum maritumum L.) grown in a cascade cropping system for short food supply chain" propose a relevant topic, namely the impact of packaging from wild arugula (main crop) and sea fennel (secondary crop) in the current context of the UN 2030 Agenda, respectively the impact on the environment of food products with a significant effect of their packaging on safety and societal security, for which we appreciate that the work has an effect of multiplication.

Citations, concepts and bibliographic references are adequately mentioned within the work, for example the bibliographic sources [20,21] are used by the authors to justify the fact that "PLA is a versatile material, being heat-sealed, a gas barrier, resistant to UV 84, biocompatible, elastic, rigid and hydrophobic", which shows us that the authors studied the specialized literature to justify their scientific arguments. Moreover, the tables and figures drawn up in the work by the authors are adequately mentioned.

The research methodology is simply presented, respectively the authors of the paper present the methods regarding the plant material and the growing conditions for the two types of crops, as well as the methods of processing, packaging and storage. Moreover, to justify the impact and effects of the use of these two plates in the packaging industry, the authors of the paper present their physico-chemical analysis, as well as the sensory analysis that "was carried out by a trained panel according to international specifications 188 (ASTM STP 913, 1986 )", as well as the realization of experimental design and statistical analysis, all these methodological elements support the application elements developed by the authors within the framework of the research results.

The results of the research presented by the authors of the paper highlight the fact that packaging freshly cut wild rocket and sea fennel with a PLA-based film is a feasible alternative to the usual plastic used in the packaging industry, avoiding the waste of polymers in landfills. aspects specific to applied research, which is why it would be useful if the authors could highlight in a separate paragraph all these innovative elements as a personal scientific contribution to the specialized literature.

The conclusions are adequately presented, the authors highlighting "a wide range of possibilities for a more ecological production and marketing of fresh cut vegetables.". However, we suggest the authors of the paper to also present what are the limitations of the study, future research being adequately presented.

We congratulate the research team for the analyzed topic, and after the review, we propose the paper for acceptance.

Author Response

The authors of the paper "Biodegradable food packaging of wild rocket (Diplotaxis tenui-folia L. [DC.]) and sea fennel (Crithmum maritumum L.) grown in a cascade cropping system for short food supply chain" propose a relevant topic, namely the impact of packaging from wild arugula (main crop) and sea fennel (secondary crop) in the current context of the UN 2030 Agenda, respectively the impact on the environment of food products with a significant effect of their packaging on safety and societal security, for which we appreciate that the work has an effect of multiplication.

Citations, concepts and bibliographic references are adequately mentioned within the work, for example the bibliographic sources [20,21] are used by the authors to justify the fact that "PLA is a versatile material, being heat-sealed, a gas barrier, resistant to UV 84, biocompatible, elastic, rigid and hydrophobic", which shows us that the authors studied the specialized literature to justify their scientific arguments. Moreover, the tables and figures drawn up in the work by the authors are adequately mentioned.

The research methodology is simply presented, respectively the authors of the paper present the methods regarding the plant material and the growing conditions for the two types of crops, as well as the methods of processing, packaging and storage. Moreover, to justify the impact and effects of the use of these two plates in the packaging industry, the authors of the paper present their physico-chemical analysis, as well as the sensory analysis that "was carried out by a trained panel according to international specifications 188 (ASTM STP 913, 1986 )", as well as the realization of experimental design and statistical analysis, all these methodological elements support the application elements developed by the authors within the framework of the research results.

The results of the research presented by the authors of the paper highlight the fact that packaging freshly cut wild rocket and sea fennel with a PLA-based film is a feasible alternative to the usual plastic used in the packaging industry, avoiding the waste of polymers in landfills. aspects specific to applied research, which is why it would be useful if the authors could highlight in a separate paragraph all these innovative elements as a personal scientific contribution to the specialized literature.

The conclusions are adequately presented, the authors highlighting "a wide range of possibilities for a more ecological production and marketing of fresh cut vegetables.". However, we suggest the authors of the paper to also present what are the limitations of the study, future research being adequately presented.

Thank you for your thorough review of the manuscript and for providing positive feedback. Based on your suggestions, we have emphasized the innovative elements as well as we have included a paragraph in the conclusions section that highlights the limitations of the study and outlines potential avenues for future research.

We congratulate the research team for the analyzed topic, and after the review, we propose the paper for acceptance.

Round 2

Reviewer 1 Report

The manuscript has been improved. The authors have addressed almost all the comments. I have only minor comments.

-Table 1 and figures 4-7. Which data belongs to day 0, day 7, OPP, and PLA? The authors rebutted they attended the comment made in the previous evaluation. However, the information shown is still unclear.   

The manuscript is well-written. The English language is fine.

Author Response

Table 1 and figures 4-7. Which data belongs to day 0, day 7, OPP, and PLA? The authors rebutted they attended the comment made in the previous evaluation. However, the information shown is still unclear.   

Thank you for your valuable assistance in improving the mentioned Table and Figures. We have implemented various changes, including the incorporation of vertical bars to enhance clarity and facilitate better understanding. We sincerely hope that these modifications have resulted in an improved presentation of the data.